# Youth Experiences with Social Norms Feedback: Qualitative Findings from The Drug Prevention Trial *the GOOD Life*

**DOI:** 10.3390/ijerph17093200

**Published:** 2020-05-04

**Authors:** Christiane Stock, Satayesh Lavasani Kjær, Birthe Marie Rasmussen, Lotte Vallentin-Holbech

**Affiliations:** 1Institute of Health and Nursing Science (IGPW), Charité—Universitätsmedizin Berlin, corporate member of Freie Universität Berlin, Humboldt-Universität zu Berlin, 13353 Berlin, Germany; 2Unit for Health Promotion Research, University of Southern Denmark, 6700 Esbjerg, Denmark; satakjaer@health.sdu.dk (S.L.K.); bmrasmussen@health.sdu.dk (B.M.R.); 3Centre for Alcohol and Drug Research, Aarhus University, 8000 Aarhus, Denmark; lvh.crf@psy.au.dk

**Keywords:** social norms, normative feedback, drug use, focus group, qualitative research, adolescents

## Abstract

*Background:* Normative feedback is an intervention strategy commonly used in drug prevention programmes. This study collected process evaluation data about how programme recipients engage with social norms (SN) feedback in *The GOOD Life* intervention and how they experience it. *Methods*: Eight focus group interviews were conducted with a total of 44 adolescents (pupils aged 14–16 years) who have participated in the social-norms-based intervention *The GOOD Life*. The interviews focused on three topics: (1) interest in and impact of the intervention; (2) perception of the intervention elements; and (3) suggestions for improvement of *The GOOD Life*. They were transcribed and analysed with content analysis. *Results*: The analysis revealed that *The GOOD Life* motivated pupils to re-evaluate their own drug use behaviour and overall met their interest regarding receiving engaging and non-moral forms of drug prevention programmes. While pupils perceived the normative feedback session in the classroom and the posters with SN messages as positive, stimulating and surprising, the web-based application with SN feedback was rarely used and less positively evaluated. Anonymity and confidentiality were regarded as essential to provide honest answers in the poll. The pupils suggested even more variety in ways to engage them and to use more gaming elements. *Conclusions*: SN feedback was well perceived by adolescents. The intervention met their interest and needs and was able to achieve the intended impact of challenging norm perceptions. Anonymity and confidentiality are key in order to build trust and engage adolescents in the intervention.

## 1. Introduction

A process evaluation is essential for informing future intervention development or for aiming at applying similar or the same intervention in other settings or populations. An important part of process evaluation is participants’ response to and interaction with the intervention in order to find the mechanisms of change and further develop the intervention [1]. Furthermore, an understanding of the challenges in implementation of complex interventions are important in order to judge why an intervention, or elements of it, do or do not work, or how the planners might expect impacts to be different if implemented in another setting or community [1]. Despite the widely acknowledged need for evaluating user experiences with prevention programmes or approaches, such knowledge is still sparse in regard to the social norms approach (SNA) [2]. SNA builds on psychological theories of behaviour, such as the theory of planned behaviour (TPB) [3], and suggests that such beliefs about peer behaviour could motivate the individual to match his or her own alcohol consumption to the perceived norm in the peer group. SNA has developed into a widely implemented behaviour change technique [4,5] and aims at correcting the often misperceived norms regarding how frequent a behaviour or attitude is among peers. Such correction of misperceived norms is mostly done through normative feedback, which is highlighting the difference between perceived versus the actual reported prevalence of the behaviour (descriptive norm) or the difference between actual and perceived peer approval of the behaviour (injunctive norm). SNA makes the assumption that interventions which correct misperceptions would promote more positive behaviours and reduce risk behaviours such as engaging in alcohol and other drug use [4].

In Denmark, regular binge drinking and drinking to get drunk is so widespread and socially accepted that it has been normalised among Danish adolescents [6]. Approximately 32% of Danes aged 15–16 years reported drunkenness in the last 30 days, while the European average for this age group is 13% [7]. Similarly, the percentage of binge drinkers in this age group, i.e., those who have consumed five or more drinks at one drinking session in the last 30 days, is high with 57% for boys and 56% for girls, respectively [7]. 

Hence, the Danish trial *The GOOD Life* aimed to investigate if the school-based SNA intervention *The GOOD Life* would be effective to correct misperceptions of peer behaviour and attitudes towards alcohol and additionally reduce alcohol use and alcohol-related harms among Danish pupils aged 13–17 years [8]. The intervention *The GOOD Life* provided tailored normative feedback for each grade at participating schools using three different elements or modes of delivery: face-to-face communication in an interactive feedback session in the classroom, posters at the school and an interactive web application. During 2015 and 2016, 1355 pupils from 38 schools were randomly assigned to intervention or control schools in the region of Southern Denmark and completed the project by answering an online survey before the intervention at baseline (T0) and 3 months after baseline (T1). The study found that pupils’ exaggerated perceptions regarding their peers’ use and approval of alcohol use were related to personal experience with alcohol [9]. It also showed that receiving the intervention had a positive effect on norm perceptions and alcohol-related harms, but a preventive effect on binge drinking was only found among pupils with an intention to drink more alcohol [10]. Hence, SNA could be a suitable preventive strategy for alcohol prevention among Danish adolescents.

Whilst many studies report the use of social normative feedback in youth alcohol prevention programmes in different settings such as schools, colleges and universities and with different modes of delivery of social norms feedback, such as media campaigns or web-based personalised feedback, only very few studies have collected information about how programme recipients actually engage with SNA feedback and how they experience it [2]. Existing research reports college students’ (dis)belief in social norms messages [11] and, more recently, perceptions of web-based SNA feedback using the Think Aloud methodology [12], but studies that either focus on how individuals respond to other forms of SNA feedback or at other age groups such as school pupils are lacking.

This qualitative research aimed to study how pupils perceived the SNA intervention *The GOOD Life* and its three elements to provide normative feedback. In particular, we aimed to answer the following questions. (1) Did the intervention meet the pupils’ interests, and did it have an impact on their norm perceptions? (2) How did pupils perceive *The GOOD Life* in general and the three intervention elements feedback sessions, posters and web application in particular? In addition (3), what kind of improvements or ideas for changes did they express? 

## 2. Materials and Methods

### 2.1. The GOOD Life Intervention

*The GOOD Life*, in Danish *Det GODE Liv*, is a social norms intervention to prevent alcohol, tobacco and marijuana use and its harmful consequences, and focuses on students aged 13–17 years. The programme challenges potential misperceptions about the magnitude of risk-taking behaviour in the reference group. This is done with realistic and positive messages about risk behaviour in the respective peer group. Details about the background and development of *The GOOD Life* are described elsewhere [5].

In brief, a baseline questionnaire measured the perceived descriptive and injunctive norms as well as the actual tobacco, alcohol, and marijuana use among the pupils. This information was used to provide normative feedback to pupils using three modes of delivery or elements of the programme corresponding to three different communication channels [5]: Face-to-face communication: The programme started at each intervention school with a 45–60 min normative feedback session conducted either in single classes or for a whole grade, enabling interaction with the pupils. This session was facilitated by a member of the research team using web-based polls, where the pupils answered questions during the session regarding perceptions, behaviour and attitudes towards alcohol, smoking and intake of marijuana, among themselves and their classmates. Their answers were compared with the results of the baseline questionnaires, and discrepancies between perceptions and factual data were discussed.Print communication: After the feedback session, teachers were asked to display 4–5 posters for each grade. The posters contained school- and grade-specific social norms messages e.g., “8 out of 10 pupils in 8th grade at [school name] have NEVER been drunk”.Interactive web application: Pupils were encouraged by teachers and by a poster in the classroom to open a web-based application on either their computer or smartphone. The poster contained a QR code, which made the access quick and easy. The pupils could receive information on the concrete behavioural norms at their school, and classes and could test their norm perceptions in a quiz. The content of social norms messages in this intervention element were similar, but not identical to the previous messages.

The posters and the web application were available for 7–8 weeks, and the entire programme lasted for approximately 8 weeks and took place at 18 intervention schools in the Region of Southern Denmark between April 2015 and October 2016.

### 2.2. Data Collection

#### 2.2.1. Focus Group Interviews

To assess the pupils’ perception of *The GOOD Life* programme, eight semi-structured focus group interviews were conducted with 2–8 participants from five different intervention schools, with a total of 44 participants aged 14–16 years (see Table 1), moderated by a female research assistant from the research team. The focus groups were conducted at all intervention schools that agreed and were able to organise focus group locations within two weeks after intervention completion. All focus groups consisted of pupils from the same school in order to ensure a certain level of homogeneity, since the participants already were familiar with each other, thus enhancing the interaction in the group [13]. The interviews were conducted in a separate room at the participants´ schools to create a safe environment for the participants. The focus groups lasted for 30 min on average and were audio recorded. 

#### 2.2.2. Participants

Pupils at intervention schools were asked to voluntarily sign up for a focus group interview via an additional question at the end of the follow-up questionnaire for the quantitative data collection (T1). A cinema voucher was offered as an incentive to participate. Using the contact information from the follow-up questionnaire, the research assistant contacted each pupil to arrange the interviews. 

After conducting the focus groups 1 and 2 in mixed groups with boys and girls (Table 1) the moderator observed that the girls tended to dominate the discussions. Therefore, the research team decided that the remaining six focus groups were to be conducted separated by sex to enable more open discussions among the boys. 

#### 2.2.3. Interview Guide

The interview guide contained discussion themes and topics about the intervention that should be elaborated and evaluated by the pupils in the interviews. Each focus group interview started by welcoming the participants and with an introduction by the moderator where the procedure and the purpose of the interview was outlined. Then, the moderator introduced herself and encouraged the participants to introduce themselves. The interview guide was semi-structured to ensure that all elements of *The GOOD Life* intervention were discussed and assessed. The questions covered the following subjects: *The GOOD Life* in general, the baseline and follow-up questionnaire, the interactive feedback session, the posters, the web application, and the influence that *The GOOD Life* had on their norm perceptions. The pupils’ responses towards subjects related to the questionnaires have not been analysed for this paper, as they do not concern the intervention itself.

### 2.3. Transcription

The interviews were audio recorded with permission from the participants and transcribed ad verbatim by two research assistants. 

### 2.4. Qualitative Data Analysis

A systematic qualitative content analysis was carried out with the aim to identify key aspects reflecting three topics, namely (1) to what extent pupils’ interests were met and their norms perceptions impacted; (2) pupils’ perceptions of *The GOOD Life*; and (3) satisfaction and suggestions for improvements from the pupils´ perspective regarding the intervention as a whole and its three SNA elements. A coding system was developed to record relevant statements, which were assigned to the three research topics above. A research assistant, who was not involved with conducting the interviews or in the data collection, coded all the material. As an analytic technique, a two-dimensional thematic matrix (cases and categories) was used, divided into the three topics to ensure a structured analysis of the research questions [14]: -Topic 1—Interest in and impact of the intervention: Quotes regarding how the intervention has met the pupils’ interest and whether it had an influence on their norm perceptions.-Topic 2—Perception of the intervention elements. The analysis schedule was divided into four subthemes: ‘general’, ‘feedback sessions’, ‘posters’, and ‘web application’. The sub-theme ‘general’ included those quotes related to the pupils’ perception towards the intervention in general and not mentioning a particular element.-Topic 3—Suggestions for improvements of *The Good Life*. Quotes regarding pupils´ suggestions for improvements of the intervention.

Within each topic, the qualitative data analysis arranged the data inductively into themes and identified patterns (interrelatedness) among the themes. Significant sub-themes were identified by what pupils said across focus groups. All topics and themes were reviewed and discussed with the last author, but not with interview partners. Thus, the last author provided feedback on the analysis based on the thematic matrix and quotes but did not participate in the analysis.

### 2.5. Ethics

Ethics approval for *The GOOD Life* trial was obtained from the Ethical Committee of the Region of Southern Denmark, and parents gave written permission for their daughter or son to participate in *The GOOD Life* trial. Participation in the focus group interviews was voluntary, and informed consent was obtained from all participants. At the beginning of each focus group, the moderator informed all participants that their interview data would be kept confidential. Pupils´ names and school names were omitted from the analysis and results.

## 3. Results

### 3.1. Topic 1: Interest in and Impact of The Intervention

The qualitative data analysis concerning this topic resulted in three sub-themes emerging from the data, namely “influence on norm perceptions”, “trustworthiness” and “anonymity and security”.

#### 3.1.1. Influence on Norm Perceptions

Most pupils expressed that their perceptions of smoking, alcohol and marijuana use among peers at their school and grade were challenged, as they generally thought that the percentages of peers engaging in the particular behaviour were a lot higher than the reported figures showed. The majority of the pupils were surprised and some rather shocked about how few pupils actually smoked, drank alcohol or used marijuana. 


*“I think it was good that we got to do this, because I think there are many—I am really surprised about it—in our class who believe that there are many who drink”.*
(boy 3, focus group 1)


*“The best thing about the project was, that it in some ways was kind of an ‘eye-opener’, since we were forced to think more about the questions being asked. For example: Have I done that? Have I not done that? And have they [the other pupils] done that?”*
(girl 2, focus group 8)

Some pupils expressed that they expect a positive impact on pupils’ decision to refuse alcohol offers.


*”I believe that there are some who think that there are not so many who drink after all. So, I believe that it is easier for them to say no to alcohol now.”*
(girl 2, focus group 2)

However, a few respondents did not share the same attitudes about the relevance of the normative feedback, and this scepticism was especially common among the boys.


*“… Well, I don’t care. I haven’t been drinking yet. My plan is to drink New Year’s Eve, but if I see that there is no one who will drink New Year’s Eve, well, that is not going to change my mind ….”*
(boy 1, focus group 2)

This opinion was not shared among girls, and some of the girls reflected that the boys did not take the issue as serious as the girls generally do as illustrated by the following quote.


*“…But, you can just see, that they [boys] think that it is fun, but it isn’t healthy. I believe that the girls think more about the issue.”*
(girl 7, focus group 5)

#### 3.1.2. Trustworthiness of The Data Used for The Social Norms Messages

Although the majority of the pupils did not express any concern about the accuracy of the data presented in the social norm messages, some pupils talked about their doubts in regard to the intervention’s validity or reflected on it.


*“The percentages that were presented didn’t quite match what I was told by the others”.*
(girl 4, focus group 1)

It was also discussed that the fact that normative feedback was provided without moral or accusing statements increased validity and honesty. 


*“Perhaps it is a good thing that we did not get an accusing finger pointing at us, since some could feel pressured to answer some things that are not true.”*
(boy 5, focus group 4)

#### 3.1.3. Anonymity and Data Security

Overall, the pupils liked the idea that during the data collection anonymity was provided. 


*“I think that it was quite interesting that you could see what the others have answered without being able to*
*know who answered what, but it was shown in overall scales.”*
(boy 2, focus group 2)

They also stated that *The GOOD Life* made them feel secure enough to answer truthfully, because anonymity was provided.


*” If it wasn’t anonymous I wouldn’t write that I smoke, if I really smoke, so I wouldn’t say it in class, but since it was anonymous on the computer, I could easily do that. So I liked that.”*
(girl 2, focus group 8)

### 3.2. Topic 2: Perceptions of The Intervention Elements

#### 3.2.1. Perceptions in General

In all focus group discussions, pupils gave generally positive feedback and expressed satisfaction with *The GOOD Life*. They expressed a positive attitude regarding the whole idea and concept of social norms messaging, where the focus is not only on their own but also on their peers’ perceptions and behaviours. Furthermore, they responded positively to the fact that the intervention did not give advice about expected behaviour (such as telling them not to use substances).


*“I really like that it is about us from this school instead of nationwide. This makes it more relevant and exiting for us, than if it was numbers for all of Denmark.”*
(boy 3, focus group 2)


*“It was also quite cool, that you shouldn’t just answer in regard to your own perception, but also about what you think how the others [the other pupils] perceive things, because then you could see the differences […] I don’t think that they [The GOOD Life intervention team] talked down to us in any way. I think, that it was a funny way of making one’s mind about this issue, since I don’t really believe that we have tried it before.”*
(girl 2, focus group 3)

Several pupils praised the fact that *The GOOD Life* is composed of several intervention elements that made the content and messages more memorable.


*“I think that it was a good way of doing it, so that we didn’t get it [the SNA messages] all at once. Compared to the other times [referring to other drug prevention programmes], where we got it all at once, and then you just end up forgetting it all faster.”*
(girl 5, focus group 5)

#### 3.2.2. Perceptions about The Feedback Session

Most of the respondents made positive statements about this intervention element because they liked the game elements and the positive engagement of the interactive feedback sessions, including web polls.


*“I also think that it was really fun, because it became a sport—You wanted to get the right answers and numbers...”*
(boy 3, focus group 2)

Several pupils praised the fact that the methods in the intervention differed from what they were used to from everyday school teaching. It made the intervention engaging and interesting, as they were actively involved. This is due to the fact that they were together with their classmates while they received surprising information about the actual results regarding drug use and approval among peers, which often resulted in laughter, common expression of surprise, and in turn elicited positive emotions of togetherness.


*“I think that it was cool—okay, so we have sat here and listened for half an hour, or something like that, but then suddenly, we should use our mobiles […] It was really fun. It was really different than just sitting and listening and listening. Then you got a break, where you should do something else. It was really cool”.*
(girl 6, focus group 7)

Meanwhile, a few pupils disagreed with the positive opinion of the other participants of the focus group, since they did not think that the feedback session was a good way of conducting an intervention:


*“… I think that there were a lot of cold hard facts all the way through. I was about to doze off […] I think, that it should have been shorter, and if it is possible, the audience should be more included.”*
(boy 7, focus group 6)

Those who were critical in regard to the feedback session wished a shorter introduction presentation and less redundancy in presenting the facts/challenging norm perceptions regarding different drug use behaviours. In addition, a higher degree of engagement was demanded to avoid boredom.

Furthermore, some pupils shared the worry of lack of confidentiality because students were sitting close to each other when responding to the web poll.


*“We sat in those groups, with whom we are also friends with, and then we sat a bit close to each other, and you could look over your shoulders and see what the others are answering. I didn’t really like that.”*
(girl 2, focus group 8)

However, this fear of being exposed was not discussed among the boys, and they did not care so much if others could watch them when responding to the web poll.


*“I couldn’t care less if there was someone who looked. I don’t think that there would be someone who would care so much if that was the case”.*
(boy 4, focus group 2)

#### 3.2.3. Perceptions about The Posters

The majority of the pupils made positive comments about the posters that were displayed at their schools, whether it was about the poster content or the design.


*“I think that it is a cool way they have chosen to illustrate it—for example the one with smoking”.*
(boy 4, focus group 4)

The pupils liked that the posters stood out and were different from other campaigns they received earlier. They stated that the posters caught their attention, which made them look and read the SNA messages on the posters. Aside from having a layout that was eye-catching, some pupils also expressed that another vital factor was where the posters were placed. 


*“…They are placed in a good spot at our school. We have one placed across our door, so when you pass through and are about to enter the classroom you can’t avoid seeing it.”*
(girl 1, focus group 1)

However, few pupils expressed that they did not care much about the posters and the messages that they contained, as they did not understand why they were put up because similar messages were already presented in the feedback session. Some participants, especially girls, discussed potential gossiping. This made them feel insecure, as they felt potentially exposed to others.


*“I can see the good things about the posters, but at the same time you are afraid that the others think that it is you who has answered this.”*
(girl 3, focus group 1)

Last but not least, some of the pupils had difficulties comprehending the content of the posters, whenever percentages were involved in the text messages. The result of this was that they did not try to read and understand the messages on posters that contained percentages due to limited mathematics skills. They would prefer that the poster messages were without these formulations. 

#### 3.2.4. Perceptions about The Web Application

With regard to the web application, pupils were encouraged to open it in a browser on a computer or their smartphone, but most of them were either unimpressed by the web application or were not aware of its existence. Only a few pupils gave some positive feedback towards the app, whereas the common criticism was that it was superfluous or not well presented to them: 


*“Well, it was quite unimportant, because I already knew the information, and we have already answered the questions three times before. I just think that it was unnecessary.”*
(girl 5, focus group 2)


*“I actually think that the reason why we didn’t remember it, was because we didn’t receive that much information about what this app was about and could do. I think that we were only told that we could download an app and what it was called—and that was it about it.”*
(boy 3, focus group 6)

### 3.3. Topic 3: Suggestions for Improvement of The GOOD Life

Furthermore, the pupils made suggestions for improvement of the intervention. They would like to be exposed to varying approaches or methods and suggested games, role playing, watching illustrative film clips, and being even more actively involved. Such elements should be combined with the elements that are already present in *The GOOD Life* intervention. Some respondents stated that too many repetitions of similar social norms messages can easily create a feeling of boredom, which makes them lose their interest and attention. Furthermore, making contests with prizes would evoke their attention.


*“Well, perhaps you could come up with something, like a game, just something that could make us get up and move around. Just a little break, or something like that […] or just use more different methods to explain it. Well, they could for example talk for about 10 min and then they could show us a movie, and then could do the web poll.”*
(girl 1, focus group 1)


*“If other schools are also involved in the intervention, perhaps a competition could be made where the price could be a Segboard, since they are popular now. That would make them download it and be ecstatic about the app, because they could win one.”*
(boy 1, focus group 6)

When comparing the discussions between focus groups, there was no substantial variation in the type of topics discussed. In addition, no substantial differences in the amount of positive or critical statements could be observed between focus groups. The only difference that became evident was the fact that boys were more open, both in terms of comments in favour of the intervention and in terms of critical remarks, in the focus groups 4 and 6 than in the mixed groups 1 and 2.

## 4. Discussion

This qualitative study aimed to investigate (1) if the SNA intervention meet the pupils’ interests and had an impact on their norm perceptions; (2) how the pupils perceived *The GOOD Life* in general and the three intervention elements feedback sessions, posters and web application in particular; and (3) which ideas for improvements they had.

With respect to the first research question, the focus group interviews showed that the social norms intervention *The GOOD Life* met the pupils´ interest and had an impact on their norm perception and motivation to refuse alcohol offers. The analysis revealed that pupils were positively impressed by and satisfied with *The GOOD Life* in general. Several quotes supported that the intervention succeeded in challenging norm perceptions and shifting perceptions about the frequency of alcohol and other drug use among peers, and some quotes indicated that the intervention made it easier for pupils to say no when offered drinks. This is in line with TPB theory suggesting that change in perceived norms is related to changes in intention for a behaviour [3]. The effect evaluation of the trial also showed that overestimation of binge drinking among peers was significantly lower in the group receiving the *The GOOD Life* than in the control group [10]. Our finding that *The GOOD Life* motivated pupils to re-evaluate their own drug use behaviour is also supported by other research. Marley et al. [12] showed that university students’ consciously engaged with a SNA intervention, which prompted them to actively consider their own behaviour, knowledge, perceptions, and to reflect on future behaviour. Further, when considering how adolescents morally regulate their behaviour and justify or excuse heavy alcohol use, peer context and anticipated social outcomes have been found to predict changes in moral disengagement related to adolescent alcohol use [15]. 

In the focus group interviews, aspects were mentioned that are likely to have contributed to the positive impact of the intervention on norm perceptions. Several pupils have stated that the intervention elements elicited positive emotions, with surprise and fun being mentioned as the most prominent emotions, which were in term related to the engaging character of the intervention. It is well documented that learning is stimulated and catalysed by emotions [16] and that interventions that engage the target group are more likely to be effective [17,18].

Our findings related to surprise elicited by *The GOOD Life* among pupils are supplemented by other research showing that students who were more surprised by their feedback also had lower estimated blood alcohol levels on their actual 21st birthday [19].

The qualitative analysis revealed also that pupils had objections in regard to traditional drug education that use moral tactics and impose adult perspectives regarding risks of drug use on them. Therefore, they appreciated that *The GOOD Life* did not use such methods and contents, but only provided feedback on data that they themselves had delivered. This is in line with others concluding that the social norms approach is well perceived by adolescents [2].

Furthermore, the respondents appreciated the whole idea and concept of social norms messaging, where the focus is not only on their own but are also on their peers’ perceptions and behaviours. This information seems to be of vital interest in this age group, where experience with alcohol and other drugs is an important characteristic of social interaction [20,21]. The pupils liked that they were exposed to something that was different from what they were used to from everyday school teaching and that they could use their smartphones to participate in the web poll. This is in line with other qualitative research showing that students willingly engaged with an online assessment and personalised feedback in regard to social norms [12]. 

Reflecting the second research question, the overall positive feedback about *The GOOD Life* was supported when the pupils were asked how they perceived the three different intervention elements: feedback session, posters and web application. It became evident that various communication channels with similar social norms messages made it easier for them to “digest” the information. However, some of the pupils would have preferred more variety during the feedback session, with e.g., including some games and showing short film clips in order to avoid boredom. Regarding the posters, the majority of the pupils liked the eye-catching design and that the content stands out from all the other educational posters that they are used to see at their school. It became evident that the location of the posters had a strong impact on their visibility and the awareness they elicit among pupils. Furthermore, some quotes indicated that there is a need for alternative ways of presenting the social messages on the posters to the pupils, as some of the pupils could not make sense of the content and meaning of the messages on their own. 

Only the web application did not achieve the intended interest among pupils and only a few in the focus groups had used it. The reasons for low use mentioned in the interviews were threefold. (1) obviously, at some schools, no information about the access to the application was provided or the pupils were not encouraged to use it. (2) Some felt that they had received enough information about peer norms through the feedback session and the posters already. (3) The web application was not interesting enough to catch their interest, because stimulating game elements such as win options were lacking. There is also some support from other research for our findings regarding the web application, as, in a trial among college students, in-person SNA feedback with a counsellor was more effective than web-based feedback [22].

Even though the focus group interviews did not support the relevance of providing social norms feedback through a web-based quiz application, we are still convinced that including this communication channel has added some value to the intervention in terms of enhancing the overall dose. Related to this, the effect analysis showed that the intervention was more effective among pupils who remembered having experienced more than one element of the intervention [23]. Therefore, we conclude both from the quantitative and the qualitative data that it is very important to deliver social norms messages though different communication channels and over an extended period of time and not only at a single occasion.

From the pupils´ perception regarding *The GOOD Life* it can also be concluded that it is very important to secure full anonymity during the entire communication process and that instructors should be aware of the physical arrangement of the chairs and tables in the room to provide anonymity when responding to a web poll or answering an online questionnaire. Other authors also stated that anonymity is important to provide high data validity, which is essential for the trustworthiness of the information provided [24]. While most of the pupils did trust the peer norm data, boys especially were more critical and more often questioned the approach as such. Despite the more critical attitudes of boys regarding *The GOOD Life*, the effect evaluation did not show any significant difference in the effectiveness of the intervention between boys and girls [25], which is in line with a review finding no evidence for sex-differences in SNA interventions among college students [26]. This indicates that social norms messages may trigger more automated decision making, even in individuals who have self-concepts of autonomous decision making and behaviour independent of peer influences.

When asked for ways to improve *The GOOD Life,* reflecting the third research question, pupils suggested to consider even more varying approaches or methods in terms of including games, role play, or showing illustrative film clips, which should all provide some active pupil involvement. Such additional methods should be combined with the elements of the present intervention. The pupils were critical about repeating social norms messages too often, which would lead to losing their interest and attention. Furthermore, including contests with prize would evoke their attention. Such contest should be considered to motivate them to download and go through the web application.

This study has limitations that need to be considered when interpreting the results. First, it explored pupil perceptions regarding delivering normative messages from only one intervention; hence, caution needs to be exercised in generalizing the findings to other SNA programmes. Second, due to self-selection of participants, the pupils in the focus groups may not represent all pupils who have experienced *The GOOD Life* and data saturation may not be fully obtained. Third, the size of the focus groups did not always meet the standard of 4–12 participants due to no show of participants. Two focus groups had only two participants, which is limiting the group discussions. Finally, the qualitative analysis was done by only one researcher, which limits its reliability. 

However, 44 adolescents interviewed provided rich, appropriate and diverse data relevant to the research question. Future research needs to explore in greater detail how participants experience other SNA interventions and SNA interventions in different settings than the school. Moreover, school context should be included in future research when studying how adolescents interact with SNA interventions.

## 5. Conclusions

Despite these limitations, our study has shed light on the under-researched field of user experiences with SNA feedback. Our study has demonstrated that, besides the overall positive user feedback, several practical considerations in relation to anonymity and confidentiality, gender differences in perceptions, and diversity of intervention methods to achieve an optimal dosage of SNA messages are important to consider when designing SNA interventions.

## Figures and Tables

**Table 1 ijerph-17-03200-t001:** Overview of the focus groups.

Focus Group	School	Girls *n*	Boys *n*	Total *n*
1	A	5	3	8
2	A	2	4	6
3	B	2		2
4	C		5	5
5	C	7		7
6	D		7	7
7	D	7		7
8	E	2		2

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
