# Peer review of "Youth Experiences with Social Norms Feedback: Qualitative Findings from The Drug Prevention Trial *the GOOD Life"

_ijerph, 2020, doi:10.3390/ijerph17093200_

Round 1

Reviewer 1 Report

The purpose of the manuscript was to provide a qualitative exploration of participant experiences of and satisfaction with a social norms intervention related to alcohol and drug use.  The paper analyzed data from 8 focus groups from 5 different schools. Findings indicate that overall participants thought the intervention was useful in challenging misperceptions about peer substance use rates.  Of all intervention elements they found the web-based application to be most effective and provided suggestions for improvement for all elements.  While this study makes a novel contribution to the literature and provides practical information for developing more effective social norms interventions there are some methodological concerns related to the analysis and the reporting of results.  Otherwise the manuscript is clearly written, with exception of some grammatical issues.  I hope the authors find my comments useful.

A couple of grammatical notes: Almost every time the word “regards” comes up in the manuscript it should actually be written as “regarding”. A search and replace would probably work, but it may also need some additional copy editing due to translation issues.  Another translation issue is the term “gaming character” which comes up several times.  A better description might be that students suggested that more “gaming elements” should be used.

Introduction

The authors should consider providing some context about substance use in Denmark, such as statistics about adolescent alcohol and drug use to establish the need for interventions that address substance use.  This would be especially important for an international audience that is not aware of this context.

Pg 1, line 36: Although this study describes one strategy used during a process evaluation it does not present a full process evaluation nor does it consider how contextual features impact the intervention to a high degree. Suggest reframing this in the introduction or focus more on the contextual features of the intervention in the analysis. See comments below for results

Pg 2, line 53: In other parts of the manuscript there is mention of the intervention also focusing on drug use.  Suggest being consistent about the type of substances the intervention addresses throughout.  Also the only drug mentioned is marijuana so if this is true instead of drugs I would just suggest saying “marijuana” (pg 3, line 95).

Pg 2, line 69: See Polonec et al. Evaluating the believability and effectiveness of the social norms message “most students  drink 0 to 4 drinks when they party as another example of a study assessing how users engage with social norms messaging.

Materials and Methods

Pg 2, line 82: Above the introduction states the program targets students ages 13-17 then in this section the intervention is described as focusing on students in 8th or 9th grade.  This does not translate well to ages that US students would be in 8th or 9th grade (13-15) and is therefore a little confusing.  Do 16-17 year olds receive different information?

Pg 2, line 91: Face to face communication- Just to clarify? this was one session only? Who delivered this information? a teacher that students were familiar with or an outside person hired to deliver the intervention?

Pg 3, line 107: For clarification at each school the intervention lasted 8 weeks? Does that just mean the posters and the interactive web application were available for 8 weeks?

Pg 3, line 112: How were the 5 schools chosen for focus groups since there were 18 total that could have been chosen?

Pg 5, line 165: Please describe this process in more detail- re- the last author discussed the codes and themes with the coder.  Why were there no attempts to have multiple researchers independently code transcripts (even a subsample) and come to a consensus about codes? If this was not done this would be a methodological limitation.  Or the authors should provide citations to support the type of analysis used if it was guided by the literature.

Results

Generally when analyzing data from focus groups there are some attempts to make comparisons across groups or to capture how interactions shaped the context of discussions. Here the results are presented more like the results came from semi-structured interviews.  While there were some attempts to discuss how gender shaped the results this could be made more explicit throughout the results section.  For example, “the theme of X came up in # out of # of the focus groups with just girls or it came up in almost all of the focus groups with girls in them but did not come up at all in the focus groups with boys.”

If the authors do want to consider how contextual factors influenced how the intervention was received it might also be interesting to add more information about the schools in table 1. For example, school size, overall socioeconomic characteristics of the school, whether the face to face intervention was delivered in classrooms or the whole school.  Then there could also be some attempts to analyze if certain themes were more prevalent in schools with certain characteristics or it could be important to mention that perhaps there were no differences across schools.  

Pg 6, line 220: Awkward wording- suggest rephrasing to something like “they responded positively to the fact that the intervention did not give advice about expected behavior (such as telling them not to use substances?)“

Pg 7, line 293: should be “unimpressed by”

Discussion

Pg 4, line 322: The authors could consider being explicit about what the focus groups found the impact to be. The only impact I saw in the results was that it challenged perceptions about peer substance use rates.

Pg 10, line 403: Another major limitation was that 2 of the focus groups only had 2 participants. An adequate size for a focus group is 4-12 with the ideal size being 5-10. 

Author Response

Reviewer 1

The purpose of the manuscript was to provide a qualitative exploration of participant experiences of and satisfaction with a social norms intervention related to alcohol and drug use.  The paper analyzed data from 8 focus groups from 5 different schools. Findings indicate that overall participants thought the intervention was useful in challenging misperceptions about peer substance use rates.  Of all intervention elements they found the web-based application to be most effective and provided suggestions for improvement for all elements.  While this study makes a novel contribution to the literature and provides practical information for developing more effective social norms interventions there are some methodological concerns related to the analysis and the reporting of results.  Otherwise the manuscript is clearly written, with exception of some grammatical issues.  I hope the authors find my comments useful.

A couple of grammatical notes: Almost every time the word “regards” comes up in the manuscript it should actually be written as “regarding”. A search and replace would probably work, but it may also need some additional copy editing due to translation issues.  Another translation issue is the term “gaming character” which comes up several times.  A better description might be that students suggested that more “gaming elements” should be used.

Response: We have edited the wording (regards – regarding) throughout the manuscript and changed “gaming character” to “gaming elements” in the abstract and in the discussion.

Introduction

The authors should consider providing some context about substance use in Denmark, such as statistics about adolescent alcohol and drug use to establish the need for interventions that address substance use.  This would be especially important for an international audience that is not aware of this context.

Response: Thank you for this suggestion. On page 2, line 62-67 we have now added the following: “In Denmark, regular binge drinking and drinking to get drunk is so widespread and socially accepted that it has been normalised among Danish adolescents [5]. Approximately 32% of Danes aged 15-16 years reported drunkenness in the last 30 days, while the European average for this age group is 13% [6]. Similarly, the percentage of binge drinkers in this age group, i.e. those who have consumed five or more drinks at one drinking session in the last 30 days, is high with 57% for boys and 56% for girls, respectively [6].”

Pg 1, line 36: Although this study describes one strategy used during a process evaluation it does not present a full process evaluation nor does it consider how contextual features impact the intervention to a high degree. Suggest reframing this in the introduction or focus more on the contextual features of the intervention in the analysis. See comments below for results

Response: Thank you for raising this important point. We have now re-phrased the first sentences of the introduction (lines 38-48) to make clear that we do not focus on contextual factors of a process evaluation, but to participants´ responses to the intervention only.

Pg 2, line 53: In other parts of the manuscript there is mention of the intervention also focusing on drug use.  Suggest being consistent about the type of substances the intervention addresses throughout.  Also the only drug mentioned is marijuana so if this is true instead of drugs I would just suggest saying “marijuana” (pg 3, line 95).

Response: In section ‘2.1 The GOOD Life intervention’ (line 100-106) we have now clarified the type of substances that were addressed in the intervention.

Pg 2, line 69: See Polonec et al. Evaluating the believability and effectiveness of the social norms message “most students drink 0 to 4 drinks when they party as another example of a study assessing how users engage with social norms messaging.

Response: We have included the suggested study on page 2, line 88; “Existing research reports college students (dis)beliefs in social norms messages (11)…”

Materials and Methods

Pg 2, line 82: Above the introduction states the program targets students ages 13-17 then in this section the intervention is described as focusing on students in 8th or 9th grade.  This does not translate well to ages that US students would be in 8th or 9th grade (13-15) and is therefore a little confusing.  Do 16-17 year olds receive different information?

Response: In Denmark, pupils in grade 8 and 9 are aged 13-15 years and 15-17 years, respectively. On page 2, lines 100-99, we have omitted the grades and specified the age-range of the students in the target group (13-17 years).

Pg 2, line 91: Face to face communication- Just to clarify? this was one session only? Who delivered this information? a teacher that students were familiar with or an outside person hired to deliver the intervention?

Response: The face-to-face communication was one session and facilitated by a member of the research team. We have clarified this on page 3, lines 111-111; “. This session was facilitated by a member of the research team using web-based polls…”

Pg 3, line 107: For clarification at each school the intervention lasted 8 weeks? Does that just mean the posters and the interactive web application were available for 8 weeks?

Response: We have clarified now in on page 3, line 126 that “The posters and the web application were available for 7-8 weeks and entire programme lasted for approximately 8 weeks..”

Pg 3, line 112: How were the 5 schools chosen for focus groups since there were 18 total that could have been chosen?

Response: To clarify this we have added a sentence on page 3, line 134-135 ‘The focus groups were conducted at all intervention schools that agreed and were able to organise focus group locations within two weeks after intervention completion. ’

Pg 5, line 165: Please describe this process in more detail- re- the last author discussed the codes and themes with the coder.  Why were there no attempts to have multiple researchers independently code transcripts (even a subsample) and come to a consensus about codes? If this was not done this would be a methodological limitation.  Or the authors should provide citations to support the type of analysis used if it was guided by the literature.

Response: We describe now the role of the last author and that she did not participate in the coding process on page 5 of the methods section “Thus, the last author provided feedback on the analysis based on the thematic matrix and quotes, but did not participate in the analysis.”

We have mentioned the fact that the coding was done by only one researcher to the limitation section in the discussion (Page 11): “Finally, the qualitative analysis was done by only one researcher which limits its validity.”

Results

Generally when analyzing data from focus groups there are some attempts to make comparisons across groups or to capture how interactions shaped the context of discussions. Here the results are presented more like the results came from semi-structured interviews.  While there were some attempts to discuss how gender shaped the results this could be made more explicit throughout the results section.  For example, “the theme of X came up in # out of # of the focus groups with just girls or it came up in almost all of the focus groups with girls in them but did not come up at all in the focus groups with boys.”

Response: We have added a section to the results section to address on comparisons across groups at page 9, lines 349-353: “When comparing the discussions between focus groups, there was no substantial variation in the type of topics discussed due to the semi-structured guidance. In addition, no substantial differences in the amount of positive or critical statements could be observed between focus groups. The only difference that became evident was the fact that boys were more open, both in terms of comments in favour of the intervention and in terms of critical remarks, in the focus groups 4 and 6 than in the mixed groups 1 and 2.”

If the authors do want to consider how contextual factors influenced how the intervention was received it might also be interesting to add more information about the schools in table 1. For example, school size, overall socioeconomic characteristics of the school, whether the face to face intervention was delivered in classrooms or the whole school.  Then there could also be some attempts to analyze if certain themes were more prevalent in schools with certain characteristics or it could be important to mention that perhaps there were no differences across schools.  

Response: We acknowledge the importance of contextual factors, but this manuscript does not have a focus on contextual factors on implementation. We have made this more clear to the reader by eliminating “context issues” from the introduction (page 1).

We suggest now in the discussion on page 11 for future research: “Moreover, school context should be included in future research when studying how adolescents interact with SNA interventions.”

Pg 6, line 220: Awkward wording- suggest rephrasing to something like “they responded positively to the fact that the intervention did not give advice about expected behavior (such as telling them not to use substances?)“

Response: We agree and have changed the wording according to your suggestion. (Page 7, lines 255-256)

Pg 7, line 293: should be “unimpressed by”

Response: Thank you for noticing, we have change “unimpressed of” to “unimpressed by”

Discussion

Pg 4, line 322: The authors could consider being explicit about what the focus groups found the impact to be. The only impact I saw in the results was that it challenged perceptions about peer substance use rates.

Response: In the discussion (page 9, lines 359-366) we have now clarified the findings on impact: “With respect to the first research question, the focus group interviews showed that the social norms intervention The GOOD Life met the pupils´ interest and had an impact on their norm perception and motivation to refuse alcohol offers. The analysis revealed that pupils were positively impressed by and satisfied with The GOOD Life in general. Some quotes indicated that the intervention made it easier for pupils to say no when offered drinks and several quotes supported that the intervention succeeded in challenging norm perceptions and shifting perceptions about the frequency of alcohol and other drug use among peers.”

We have also added a sentence to the results section to point out the alcohol refusal impact indicated by empirical quotes: “Some pupils expressed that they expect a positive impact on pupils´ decision to refuse alcohol offers”. (page 5, line 216)

Pg 10, line 403: Another major limitation was that 2 of the focus groups only had 2 participants. An adequate size for a focus group is 4-12 with the ideal size being 5-10. 

Response: We have now addressed this limitation in the limitation section on page 11, lines 451-453. “Third, the size of the focus groups did not always meet the standard of 4-12 participants due to no show of participants. Two focus groups had only two participants, which is limiting the group discussions.”

Reviewer 2 Report

1. The introduction should be expanded by adding one or more theoretical models useful for understanding the results.

2. The author(s) should motivate why the interviews were not analyzed using statistical software.

3. The results may be interpreted/discussed using the role of moral disengagement mechanisms (see D’Urso, G., Petruccelli, I., & Pace, U. (2018). Drug use as risk factor of Moral Disengagement: a study on drug traffickers and offenders against other persons. Psychiatry, Psychology and Law, 25(3), 417-424. doi:10.1080/13218719.2018.1437092).

The author(s) should create a section with limitations and future prospectives of this study.

Author Response

Reviewer 2

  1. The introduction should be expanded by adding one or more theoretical models useful for understanding the results.

Response: Thank you for this comment. We have introduced the Theory of Planned Behaviour (TPB) in the introduction (page 2, line 51-53) as a theory relevant to explain the impact of the intervention on norm perceptions as results of the interviews. We discuss the results in light of TPB on page 9, lines 365-366.

  1. The author(s) should motivate why the interviews were not analyzed using statistical software.

Response: We did not use statistical software as the focus of the qualitative analysis was not on statistical comparisons. Software for the analysis of qualitative data was not used, because quantifying of quotes was not desired.

  1. The results may be interpreted/discussed using the role of moral disengagement mechanisms (see D’Urso, G., Petruccelli, I., & Pace, U. (2018). Drug use as risk factor of Moral Disengagement: a study on drug traffickers and offenders against other persons. Psychiatry, Psychology and Law, 25(3), 417-424. doi:10.1080/13218719.2018.1437092).

Response: We have now discussed the results in the light of moral disengagement mechanisms and have added the statement to the discussion on page 9-10, lines 372-374: “Further, when considering how adolescents morally regulate their behaviour and justify or excuse heavy alcohol use, peer context and anticipated social outcomes have been found to predict changes in moral disengagement related to adolescent alcohol use (15)”.

The author(s) should create a section with limitations and future prospectives of this study.

Response: We have added/revised a limitations section and have added future prospectives to the discussion on page 11, lines 446-454:

“This study has limitations that need to be considered when interpreting the results. First, it explored pupil perceptions regarding delivering normative messages from only one intervention, hence caution needs to be exercised in generalizing the findings to other SNA programmes. Second, due to self-selection of participants, the pupils in the focus groups may not represent all pupils who have experienced The GOOD Life and data saturation may not be fully obtained. Third, the size of the focus groups did not always meet the standard of 4-12 participants due to no show of participants, and two focus groups had only two participants, which is limiting the group discussions. Finally, the qualitative analysis was done by only one researcher which limits its reliability. However, 44 adolescents interviewed provided rich, appropriate and diverse data relevant to the research question. Future research needs to explore in greater detail how participants experience other SNA interventions and SNA interventions in different settings than the school. Moreover, school context should be included in future research when studying how adolescents interact with SNA interventions.”

Round 2

Reviewer 1 Report

The authors have addressed all of my comments and concerns.